# Evaluation of Helium Ion Radiotherapy in Combination with Gemcitabine in Pancreatic Cancer In Vitro

**DOI:** 10.3390/cancers16081497

**Published:** 2024-04-14

**Authors:** Bahar Cepni, Thomas Tessonnier, Ivana Dokic, Stephan Brons, Bouchra Tawk, Andrea Mairani, Amir Abdollahi, Jürgen Debus, Klaus Herfarth, Jakob Liermann

**Affiliations:** 1Heidelberg University School of Medicine, 69120 Heidelberg, Germany; baharcepni@gmail.com; 2Heidelberg Ion-Beam Therapy Center (HIT), Department of Radiation Oncology, Heidelberg University Hospital, 69120 Heidelberg, Germany; thomas.tessonnier@med.uni-heidelberg.de (T.T.); stephan.brons@med.uni-heidelberg.de (S.B.); andrea.mairani@med.uni-heidelberg.de (A.M.); juergen.debus@med.uni-heidelberg.de (J.D.); klaus.herfarth@med.uni-heidelberg.de (K.H.); 3Clinical Cooperation Unit Translational Radiation Oncology, German Cancer Consortium (DKTK) Core-Center Heidelberg, National Center for Tumor Diseases (NCT), Heidelberg University Hospital (UKHD) and German Cancer Research Center (DKFZ), 69120 Heidelberg, Germany; ivana.dokic@med.uni-heidelberg.de (I.D.); bouchra.tawk@med.uni-heidelberg.de (B.T.); amir.abdollahi@med.uni-heidelberg.de (A.A.); 4Division of Molecular and Translational Radiation Oncology, Heidelberg Faculty of Medicine (MFHD) and Heidelberg University Hospital (UKHD), Heidelberg Ion-Beam Therapy Center (HIT), 69120 Heidelberg, Germany; 5Heidelberg Institute of Radiation Oncology (HIRO), National Center for Radiation Oncology (NCRO), Heidelberg University Hospital and German Cancer Research Center (DKFZ), 69120 Heidelberg, Germany; 6National Center of Radiation Oncology, Heidelberg Institute of Radiation Oncology, 69120 Heidelberg, Germany

**Keywords:** helium ion radiotherapy, pancreatic cancer, gemcitabine, RBE, modified microdosimetric kinetic model (mMKM)

## Abstract

**Simple Summary:**

Pancreatic cancer is one of the most aggressive cancers. New treatment strategies such as particle radiotherapy could offer a way to overcome the limitations in treatment caused by the characteristics of pancreatic cancer. Helium ions represent an attractive therapy option because of their physical and radiobiological features. The aim of this study is to investigate the efficacy of helium ion irradiation in pancreatic cancer cell lines and whether a combination with chemotherapy could increase its efficacy. The data generated in this study may serve as the radiobiological basis for future experimental and clinical works using helium ion radiotherapy in pancreatic cancer treatment.

**Abstract:**

Background: Pancreatic cancer is one of the most aggressive and lethal cancers. New treatment strategies are highly warranted. Particle radiotherapy could offer a way to overcome the radioresistant nature of pancreatic cancer because of its biological and physical characteristics. Within particles, helium ions represent an attractive therapy option to achieve the highest possible conformity while at the same time protecting the surrounding normal tissue. The aim of this study was to evaluate the cytotoxic efficacy of helium ion irradiation in pancreatic cancer in vitro. Methods: Human pancreatic cancer cell lines AsPC-1, BxPC-3 and Panc-1 were irradiated with photons and helium ions at various doses and treated with gemcitabine. Photon irradiation was performed with a biological cabin X-ray irradiator, and helium ion irradiation was performed with a spread-out Bragg peak using the raster scanning technique at the Heidelberg Ion Beam Therapy Center (HIT). The cytotoxic effect on pancreatic cancer cells was measured with clonogenic survival. The survival curves were compared to the predicted curves that were calculated via the modified microdosimetric kinetic model (mMKM). Results: The experimental relative biological effectiveness (RBE) of helium ion irradiation ranged from 1.0 to 1.7. The predicted survival curves obtained via mMKM calculations matched the experimental survival curves. Mainly additive cytotoxic effects were observed for the cell lines AsPC-1, BxPC-3 and Panc-1. Conclusion: Our results demonstrate the cytotoxic efficacy of helium ion radiotherapy in pancreatic cancer in vitro as well as the capability of mMKM calculation and its value for biological plan optimization in helium ion therapy for pancreatic cancer. A combined treatment of helium irradiation and chemotherapy with gemcitabine leads to mainly additive cytotoxic effects in pancreatic cancer cell lines. The data generated in this study may serve as the radiobiological basis for future experimental and clinical works using helium ion radiotherapy in pancreatic cancer treatment.

## 1. Introduction

Pancreatic cancer is amongst the seven leading causes of cancer-caused deaths [1]. It is one of the most aggressive and lethal cancers with a 5-year overall survival (OS) rate of 10% [2,3]. Surgery is the only potential cure; however, only 15–20% of the patients are suitable for resection because of late diagnosis. Even after R0 resection, the prognosis is poor [4].

Chemotherapy and chemoradiation are therapeutic options applied in neoadjuvant or adjuvant concepts, whereas the stage of the disease is the decision parameter for the appropriate therapy. It has been observed in numerous studies, that neoadjuvant therapy with FOLFIRINOX or gemcitabine can convert irresectable tumors into resectable ones in patients with locally advanced pancreatic cancer [5,6]. Chemoradiation leads to a higher local tumor control in pancreatic cancer but so far, an undoubtful improvement in the OS rate has not been demonstrated [7,8].

Pancreatic cancer has two features limiting the efficacy of radiotherapy: it is relatively radioresistant [9] and the surrounding organs are highly radiosensitive. Particle radiotherapy could offer a way to overcome these limitations because of its biological and physical characteristics. The maximum dose deposition occurs at the Bragg peak at a defined depth in the target tissue [10]. Biologically, particle radiation generates complex DNA lesions via its high linear energy transfer (LET) and has a higher RBE than conventional photon therapy [11,12].

Helium ions have a sharp penumbra because of their lower lateral scattering [13]. Being heavier than protons but lighter than carbon ions, helium ions represent an attractive therapy option to achieve the highest possible conformity while producing less damage than heavy ions in the surrounding normal tissue. The first data on helium ion irradiation in pancreatic cancer came from the Lawrence Berkeley Laboratory in California (USA) in the 1980s. The postmortem examination of 22 patients showed the safety of helium ion irradiation [14]. In addition, Saunders and Castro irradiated uveal melanoma patients with helium ions, which led to a local tumor control rate of 97% [15,16]. To date, helium ions are rarely used in clinical settings. Initial studies at the Heidelberg Ion Beam Therapy Center (HIT) showed that helium ion therapy could offer better organs-at-risk sparing compared to protons or conventional photon radiotherapy [17] for head and left-sided breast cancers. Further work is investigating the potential of helium ions for other indications, such as pancreatic cancer.

The aim of the current study is to compare the effect of helium ion irradiation with photon irradiation in pancreatic cancer cell lines and to evaluate the effect of combined chemotherapy and helium ion radiation compared to the conventional gemcitabine-based chemoradiation. Furthermore, the experimentally estimated clonogenic survival curves were compared to predicted curves that were calculated applying the modified microdosimetric kinetic model (mMKM), developed for biological plan optimization at our institution [18] and implemented in the clinical treatment planning system (TPS) RayStation (Raysearch Labs, Stockholm, Sweden).

## 2. Materials and Methods

### 2.1. Cell Culture and Reagents

The pancreatic cancer cell lines AsPC-1, BxPC-3 and Panc-1 were obtained via the American Type Culture Collection (ATCC, Manassas, VA, USA). The cell lines had a comparable doubling time but different mutations, tumorigenicity levels, angiogenic potentials and adhesive abilities [19,20,21]. Thus, the selected cell lines are thought to be representative of pancreatic cancer in vivo.

For the cultivation of the cell lines BxPC-3 and Panc-1, RPMI1640 medium was used, whereas AsPC-1 cells were cultivated in D-MEM medium. Together, 10% fetal bovine serum (FBS) and 1% penicillin–streptomycin were added to all media. Cells were kept at 37 °C with 5% CO_2_ and 95% humidity.

### 2.2. Photon Radiotherapy

Photon irradiation was applied at room temperature with a biological cabinet X-ray irradiator (XRAD 320 Precision X-ray Inc., N. Bradford, CT, USA) at single doses of 2, 4 and 6 Gy (gray).

### 2.3. Helium Ion Radiotherapy

Helium ion radiotherapy was applied at the HIT with a horizontal beamline using the raster scanning technique. Single doses of 1, 2 and 3 Gy were delivered, using a 3 cm acrylic shield to adjust the position of the cell monolayers in the middle of the extended Bragg peak. The planned spread-out Bragg peak (SOBP) is 1 cm in depth, centered at 3.5 cm water equivalent. The LET at the cells’ position is approximately 19.4 keV/µm. Non-irradiated cells were used as controls.

### 2.4. Gemcitabine Therapy

Gemcitabine (HEXAL, Holzkirchen, Germany) was applied at different concentrations in the cell media. Cells were exposed to this treatment for 4 h before media change and irradiation. Single doses of 10, 30, 50 and 70 nM were applied.

### 2.5. Clonogenic Assay

To evaluate the efficacy of the applied therapy, clonogenic survival assays were performed. Before the treatment, a defined number of cells (200–10,000 cells, depending on the dosage to be applied) were seeded into 25 cm^2^ flasks (Greiner Bio-One GmbH, Frickenhausen, Germany). After cell adhesion (~24 h), the treatment was performed. Following the treatment, the flasks were incubated for 8–14 days, depending on the cell line, until the surviving cells formed colonies. The cells were covered with a solution of methanol and acetic acid (3:1) for 10 min for fixation, followed by an application of 0.1% crystal violet for 10 min for staining. The colonies formed by surviving cells were counted under the microscope, considering only colonies with a minimum of 50 cells to be surviving. Surviving fractions were thus determined and were used to calculate the plating efficiency [22] and clonogenic survival, from which the clonogenic survival curves were generated. α- and β-parameters were determined and the RBE was calculated. Each experiment was repeated independently, using triplicates of each dose. Linear quadratic fits were performed, and surviving fraction data were plotted as mean values and standard deviation determined by single-treatment experiments for each cell line. The results are presented as mean values and standard deviation.

### 2.6. Combined Chemo- and Radiotherapy

Two doses of each therapy modality were chosen and combined for combination experiments, which achieved moderate and severe cell death. Doses of 2, 4 and 6 Gy of photon irradiation, 1, 2 and 3 Gy of helium ion irradiation and 10 and 50 nM of gemcitabine were used. Surviving fractions were calculated for the cell lines after the combination treatment, which were then normalized to a drug control sample that was treated with only the respective dose of gemcitabine. To normalize the surviving fraction, the ratio of the plating efficiency of the combined therapy to the plating efficiency of the respective dose of gemcitabine monotherapy was calculated.

The normalized fractions were then compared with another control sample that was only treated with photons or helium ions without the application of gemcitabine. The surviving fractions after the mono radiotherapy with photons or helium ions are shown in the control curve. Results in the area under the control curve were evaluated as additive cytotoxic and results within the control curve as independent. Synergistic effects were evaluated according to the established model by Steel and Peckham [23].

### 2.7. RBE-Weighted Dose Calculation with mMKM

The RBE-weighted dose calculations were performed in the TPS using an implementation of the mMKM as previously described [18]. Dedicated mMKM biological tables were created for each cell line using the α- and β-parameters determined from the survival data of the low-energy photon irradiation. This dataset was corrected to take into account the higher efficiency of low-energy photons compared to high-energy photons (about 1.05) [24]. The dose-dependent RBE values and mMKM-predicted cell survival curves were estimated using the irradiated plans at different dose levels.

## 3. Results

### 3.1. Clonogenic Survival and RBE after Photon versus Helium Irradiation and Gemcitabine Treatment

A dose-dependent suppression of the survival fraction was observed in every cell line after both photon and helium ion therapy. The survival fractions after helium ion treatment were lower than those after photon treatment at the same physical dose. Helium ion radiotherapy led to an enhanced cell killing in the cell lines AsPC-1, BxPC-3 and Panc-1 compared to the conventional photon radiotherapy (Figure 1). The effect of helium ion radiotherapy was similar in the cell lines BxPC-3 and Panc-1, with a decrease in survival fraction (SF) after the maximal dose of 3 Gy helium irradiation to a minimum of 14% in BxPC-3 and 12% in Panc-1. The SF values after 1 and 2 Gy helium irradiation were also corresponding (1 Gy-Panc1-SF: 57%, 1 Gy-BxPC3-SF: 56%; 2 Gy-Panc1-SF: 28%, 2 Gy-BxPC3-SF: 25%). AsPC-1 had the highest sensibility to helium ion radiotherapy with an alpha/beta-ratio of 0.12 Gy and a minimal SF of 8% after the maximal radiation dose of 3 Gy.

Based on the experimental results, we observed RBE values for helium from 1.0 to 1.7, depending on the radiation dose and the cell line (Table 1).

The mMKM-estimated RBE for helium ion radiotherapy was determined from the mMKM-estimated linear quadratic fits that were performed and achieved RBE values from 1.4 to 1.7 for all cell lines (Table 2).

The mean RBE value differences between the experimental data and the mMKM estimations were for AsPC-1, 0.79 at 1 Gy, 0.16 at 2 Gy and 0.19 at 3 Gy; for BxPC-3, 0.16 at 1 Gy, 0.10 at 2 Gy and 0.01 at 3 Gy and for Panc-1, 0.22 at 1 Gy, 0.28 at 2 Gy and 0.32 at 3 Gy. The overall range of difference was 3.7 ± 19.8%. The mMKM calculation-predicted cell survival was overall largely matching to the experimentally determined values in the cell lines AsPC-1 and BxPC-3, whereas Panc-1 had a slight discrepancy (Figure 2).

Gemcitabine therapy also led to a dose-dependent suppression of cell survival, proving Panc-1 to be the most resistant to gemcitabine treatment (Figure 3). The SF for Panc-1 was 50% after 50 nM of gemcitabine therapy. The cell line AsPC-1 showed the highest sensibility to gemcitabine therapy and had the steepest decrease in the survival curve.

### 3.2. Cytotoxic Effect of Combined Chemo- and Radiotherapy

Combined photon treatment with gemcitabine showed mainly additive effects in all three cell lines (Figure 4). Supra-additivity as defined by Steel and Peckham could not be observed. AsPC-1 had an SF of 60% after 2 Gy photon radiotherapy and 50 nM gemcitabine compared to the control curve, whereas the SF after the combined therapy of 2 Gy photon radiotherapy and 10 nM of gemcitabine therapy was 90% of the control value. After 2 Gy photon radiotherapy and 50 nM gemcitabine therapy, BxPC-3 showed a decrease of 50% in SF compared to the control. The values of Panc-1 were similar to the control curve.

Combined helium irradiation with gemcitabine also led to mainly additive effects in all of the cell lines (Figure 5). AsPC-1 showed a comparable SF after 10 nM and 50 nM of combined gemcitabine therapy. BxPC-3 also demonstrated similar results after both doses of combined gemcitabine therapy. Only Panc-1 showed a higher decrease in SF after the higher dose of 50 nM combined gemcitabine therapy, compared to the 10 nM gemcitabine therapy.

## 4. Discussion

In this study, we evaluated the cytotoxic effects of helium ion irradiation with and without the combination of gemcitabine-based chemotherapy in pancreatic cancer in vitro. In the second step, we compared the results with the effect of the conventional photon irradiation and gemcitabine-based chemoradiation. It was the aim of this study to evaluate a new radiotherapeutic technique, which could improve the so far limited therapeutic success of photon irradiation.

An RBE between that of protons and heavy ions was to be expected, due to the intermediate weight of helium ions, being heavier than protons but lighter than carbon ions. Experimental data showed an RBE of 1.1 for protons [25] and an RBE of 2–3 for carbon ions [26]. The experimental data from the Berkeley laboratory showed RBE values for helium ions ranging from 1.2 to 1.4 [15]. The observed RBE values in this study (1.0–1.7) are, thus, compatible with those data. RBE models are highly complex, integrating several physical and biological parameters for accurate calculations. In addition to the potential inaccuracies caused by the RBE model, there are systematic inaccuracies between in vivo and in vitro experiments. Thus, the expansion of the data and the optimization of the RBE models are very important for the clinical use of helium ions [24].

The mMKM was chosen for the first clinical helium irradiation at HIT [18]; therefore, it was the RBE model chosen to be used in this study. The cell survival calculated by the mMKM model was mostly in accordance with the experimental data. BxPC-3 showed the lowest difference in mean RBE values (0.01–0.16), followed by Panc-1 (0.22–0.32). AsPC-1 had the highest difference (0.79–0.16). It should be considered that experimental values have potential errors. For example, the experimental RBE value for AsPC-1 at 1 Gy was lower than 1, most likely caused by experimental error. On the other side, mMKM parameters are based on experimental data; therefore, the linear quadratic fits also include potential errors. Considering the potential errors, the results showed that the mMKM model seems to be an adequate RBE model for helium ion irradiation.

The results of gemcitabine treatment proved Panc-1 to be gemcitabine-resistant compared to the other cell lines, which has been described in the literature [27]. In accordance with this observation, Panc-1 cells have shown slight additive effects after combined therapy with gemcitabine. Considering the literature, this study is the first to examine the combination of helium ion irradiation and gemcitabine in pancreatic cancer in vitro. El Shafie et al. observed additive effects on the same three cell lines after treating them with carbon ion irradiation and gemcitabine [28]. The result of additive effects should be interpreted carefully, since the treatment had high cytotoxicity (especially for the 50 nM Gemcitabine doses), leading to low survival and relatively large statistical uncertainties. In addition, it has been shown that the cytotoxicity of particle therapy is less dependent on the cell cycle than photon therapy [29], producing more DNA double-strand breaks (DBSs) [30], whereas the radiosensitizing effect of gemcitabine is explained by the synchronization of the cell cycle at the radiosensitive G1/S junction [31]. Since the main molecular mechanisms of both treatment modalities seem to be independent, additive effects are caused. Mainly additive effects were achieved in all of the cell lines after the combined treatment with helium ions and gemcitabine, with the effect on the cell line Panc-1 being the lowest.

Although gemcitabine has been part of the standard therapy of advanced pancreatic cancer for decades, nearly all tumors develop some kind of gemcitabine resistance. In order to find a strategy to overcome this resistance, one phase III randomized controlled trial investigated the combination of gemcitabine with the selective EGFR inhibitor Erlotinib, which improved the overall median survival by less than two weeks [32]. Recent clinical trials investigate small molecule inhibitors to enhance treatment options for pancreatic cancer. To date, disease treatment with FOLFIRINOX and variations on gemcitabine combination therapy showed modest improvements in patient survival [33]. Current data demonstrated that FAK inhibition can render previously unresponsive pancreatic tumors responsive to chemo- and immunotherapy [34]. Other trials investigated ways to inhibit cell pathways resulting in cancer aggressiveness and resistance. The newly developed 3-amino-1,2,4-triazines, targeting PDK, showed promising therapeutic potential for combatting highly aggressive KRAS-mutant pancreatic ductal adenocarcinoma. It was demonstrated that PDK inhibition had equal efficacy and a better tolerability profile compared to cisplatin and gemcitabine [35].

## 5. Conclusions

The data generated by this study will serve as the basis for future experimental and clinical works, being the first to investigate the effect of helium ion irradiation on pancreatic cancer cell lines in vitro. Our results show the capability of mMKM calculations to describe in vitro data and their value for biological plan optimization in helium ion therapy. Thus, the present work lays the foundation for the clinical use of helium ion radiation to treat pancreatic cancer patients in the near future.

## Figures and Tables

**Figure 1 cancers-16-01497-f001:**
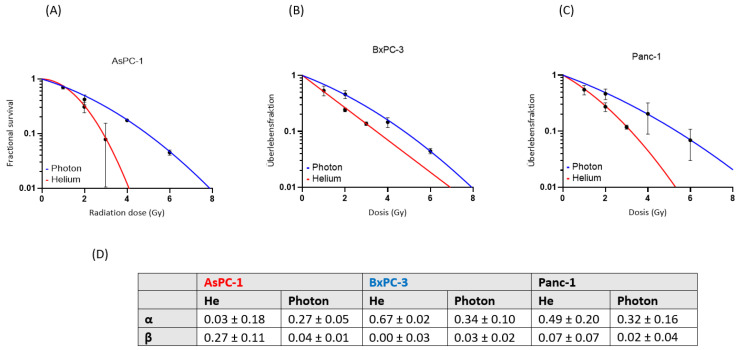
Single-treatment survival curves for AsPC-1, (**A**) BxPC-3 (**B**) and Panc-1 (**C**). Fractional survival (red: helium ion, blue: photon) is shown in dependence on the radiation dose (Gy). The α and β values are shown (mean and standard deviation) for each cell line (**D**).

**Figure 2 cancers-16-01497-f002:**
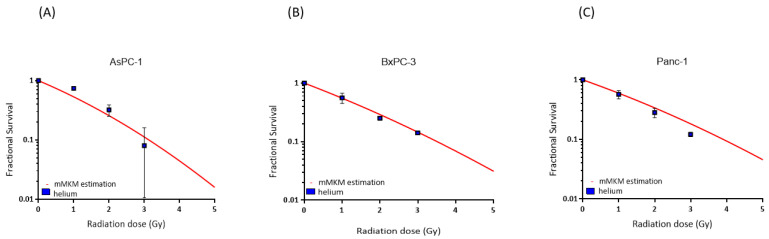
mMKM calculation for cells AsPC-1, (**A**) BxPC-3 (**B**) and Panc-1 (**C**). The mMKM-predicted fractional survival is illustrated as red curves for each cell line as a function of the radiation dose (Gy). The experimental helium ion survival is shown in blue squares.

**Figure 3 cancers-16-01497-f003:**
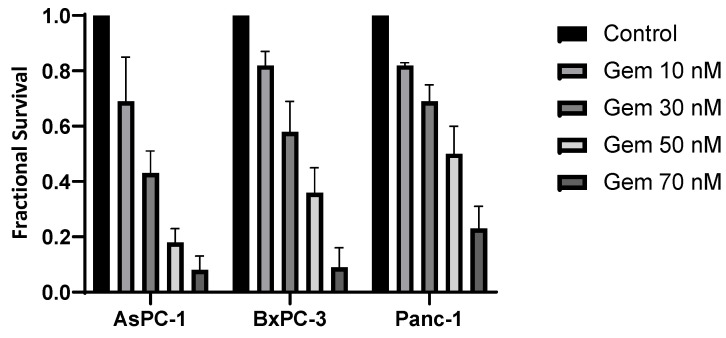
Surviving fractions after treatment with gemcitabine (Gem). Fractional survival of each cell line is shown in dependence on the gemcitabine dose.

**Figure 4 cancers-16-01497-f004:**
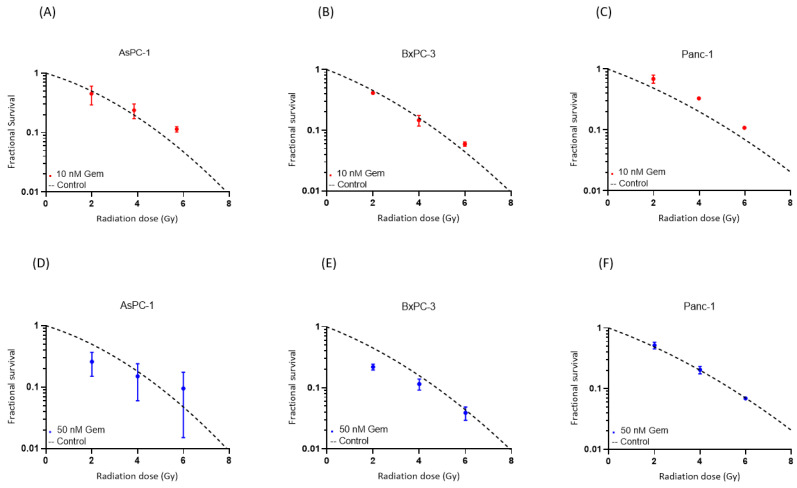
Survival after the combination of gemcitabine treatment with photon irradiation. Experimental survival fraction data are shown as dots (means with standard deviation) in comparison to the control curves for each cell line. (Appendix A).

**Figure 5 cancers-16-01497-f005:**
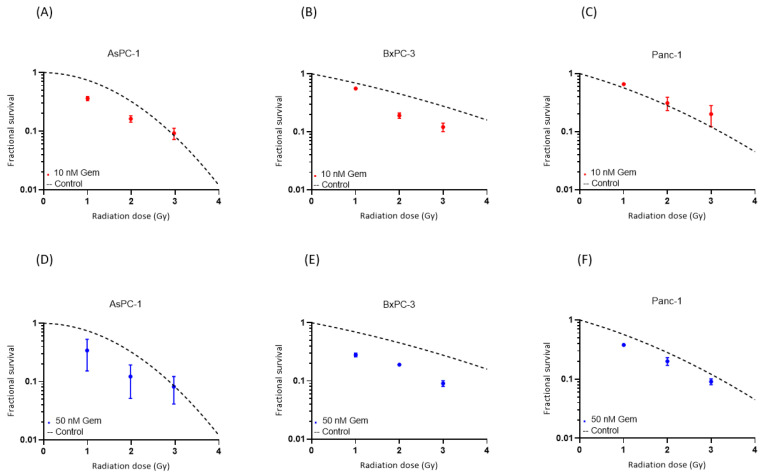
Survival after the combination of gemcitabine treatment with helium ion irradiation. Experimental survival fraction data are shown as dots (means with standard deviation) in comparison to the control curves for each cell line. (Appendix A).

**Table 1 cancers-16-01497-t001:** Experimental RBE values for helium from 1.0 to 1.7, depending on the radiation dose and the cell line.

Experimental	RBE		
Radiation Dose	AsPC-1	BxPC-3	Panc-1
**1 Gy**	0.96	1.71	1.65
**2 Gy**	1.44	1.58	1.66
**3 Gy**	1.69	1.40	1.67

**Table 2 cancers-16-01497-t002:** mMKM-estimated RBE values for helium from 1.4 to 1.7, depending on the radiation dose and the cell line.

mMKM	RBE		
Radiation Dose	AsPC-1	BxPC-3	Panc-1
**1 Gy**	1.75	1.55	1.43
**2 Gy**	1.60	1.48	1.38
**3 Gy**	1.50	1.41	1.35

## Data Availability

The original contributions presented in the study are included in the article; further inquiries can be directed to the corresponding author.

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
