# Peer review of "Evaluation of Helium Ion Radiotherapy in Combination with Gemcitabine in Pancreatic Cancer In Vitro"

_cancers, 2024, doi:10.3390/cancers16081497_

Round 1

Reviewer 1 Report

Comments and Suggestions for Authors

The manuscript was aimed to examine the cytotoxic effects of helium ion irradiation with photon irradiation in pancreatic cancer cell lines and further evaluate the effects of chemotherapy (e.g., gemcitabine) and helium ion radiation compared with gemcitabine-based chemoradiation.

I have the following concerns and suggestions about this manuscript:

1) It will be better to show IC50 values for gemcitabine for 3 pancreatic cell lines used in present study and further compare these values for cancer cells treated with chemotherapy combined with helium ion radiotherapy. The authors have to include fold-difference ration into the Table to show the benefits of combined therapies. 

2) to deliniate between additive and synergistic effects of treatments indicated above the authors have to use the appropriate tools (e.g. Synergy Finder, etc.) to get clear evidence whether helium ion radiotherapy has the additive or synergistic effects on the chemo-treated cancer cells. After this point will be addressed properly, the authors have to re-phrase the sentences like "combined photon treatment with gemcitabin showed MAINLY additive effects" (lines 206), or "combined helium irradiation with gemcitabin led also to MAINLY additive effects" (line 216). 

3) It is highly desirable to examine the mechanisms of cytotoxic activities of the single and combined treatments of pancreatic cancer cell lines. For this purpose, the authors might use western blotting to examine expression of the apoptotic markers (e.g. cleaved PARP and/or caspase-3), or flow cytometry to count the number of Annexin V-positive cells. 

Reviewer 2 Report

Comments and Suggestions for Authors

Query#1:

The introduction lacks depth in discussing existing therapies for pancreatic cancer. It would be beneficial for the authors to expand on this topic, particularly by incorporating insights from recent literature. I suggest citing the following updated references to enhance the discussion:

Principe, D. R., Underwood, P. W., Korc, M., Trevino, J. G., Munshi, H. G., & Rana, A. (2021). The Current Treatment Paradigm for Pancreatic Ductal Adenocarcinoma and Barriers to Therapeutic Efficacy. Frontiers in Oncology, 11, 688377. DOI: 10.3389/fonc.2021.688377

Sun, J., Russell, C. C., Scarlett, C. J., & McCluskey, A. (2020). Small molecule inhibitors in pancreatic cancer. RSC Medicinal Chemistry, 11(2), 164–183. DOI: 10.1039/c9md00447eI

 Jiang, H., Hegde, S., Knolhoff, B. L., Zhu, Y., Herndon, J. M., Meyer, M. A., ... DeNardo, D. G. (2016). Targeting focal adhesion kinase renders pancreatic cancers responsive to checkpoint immunotherapy. Nature Medicine, 22(8), 851–860. DOI: 10.1038/nm.4123

 Carbone, D., De Franco, M., Pecoraro, C., Bassani, D., Pavan, M., Cascioferro, S., ... Diana, P. (2023). Discovery of the 3-amino-1,2,4-triazine-based library as selective PDK1 inhibitors with therapeutic potential in highly aggressive pancreatic ductal adenocarcinoma. International Journal of Molecular Sciences, 24(4), 3679. DOI: 10.3390/ijms24043679

Pecoraro, C., De Franco, M., Carbone, D., Bassani, D., Pavan, M., Cascioferro, S., Parrino, B., Cirrincione, G., Dall'Acqua, S., Moro, S., Gandin, V., & Diana, P. (2023). 1,2,4-Amino-triazine derivatives as pyruvate dehydrogenase kinase inhibitors: Synthesis and pharmacological evaluation. European journal of medicinal chemistry, 249, 115134.

Query#2:

Provide a more comprehensive explanation in the "Introduction" regarding the selection criteria for the cell lines (AsPC-1, BxPC-3, and Panc-1) used in the study.

General Comment:

Ensure thorough proofreading of the entire document to rectify typographical errors. 

These revisions will enrich the discussion and improve the overall quality of the article, enhancing its potential contribution to the scientific literature on pancreatic cancer treatments.

Comments on the Quality of English Language

The article titled "Evaluation of Helium Ion Radiotherapy in Combination with Gemcitabine in Pancreatic Cancer In Vitro" explores the pharmacological evaluation of a novel therapeutic approach for pancreatic cancer, involving radiotherapy irradiation with photons and helium ions at various doses as adjuvant therapy with gemcitabine.

This article holds promise for making a substantial contribution to the scientific literature on pancreatic cancer treatments. However, I suggest that the authors consider making some revisions before publication.

Reviewer 3 Report

Comments and Suggestions for Authors

2. Materials and Methods

Why is the culture medium used different even though you purchased the cell lines from ATCC?

BxPC-3

https://www.atcc.org/products/crl-1687#detailed-product-information

RPMI1640 is correct.

AsPC-1

https://www.atcc.org/products/crl-1682#detailed-product-information

RPMI1640 is correct.

Panc-1

https://www.atcc.org/products/crl-1469#detailed-product-information

D-MEM is correct. 

Line 106: 

CO2 → CO2

2.4. Gemcitabine Therapy

Which maker did you purchase Gemcitabine?

Line 125:

25 cm2 → 25 cm2

Line 124: 

24hr or 24 hours?

Figure1:

Please make the three graphs the same size.

Line 190:

3.7% +/- 19.8%  ã€€3.7±19.8 %

Figure 2:  

Please make the three graphs the same size.

Figure 3: 

Please show the footnote does not appear as Gem, not use GEM or change the notation in the figure to GEM.

Figure 4: 

The thickness of the axes of the graph are different, so please align them.

Figure 5: 

The thickness of the axes of the graph are different, so please align them.

*It is very important to have the same size, height, and font throughout the figures to make it easier for the reader to see.

Discussion:

Too many line breaks. Please organize your paragraphs according to their content.

Line 273: 

In vitro  ã€€in vitro

Conclusion: 

I understand the importance of your data, but I think it would be better to be more specific about how to use your data to emphasize its importance.

References:

Some links are broken, others are included, etc. Please unify.

Even at a quick glance, your self-citations exceed 20 %, so I think you should add some references from a different groups.

Round 2

Reviewer 1 Report

Comments and Suggestions for Authors

The authors responded to my suggestions and concerns.

As I mentioned previously,  the manuscript is a bit descriptional.  The authors agreed with this point in their answer 3 and highlighted that "our work is just scratching on the surface of mechanistic explanation" and in further studies they are planing  to examine the underlying mechanisms. Thus, I recommend to accept this manuscript for publication as a communication. 

Reviewer 3 Report

Comments and Suggestions for Authors

It is understandable that you cited yourselves because your works are important, but if the self-citation rate is too high, it becomes difficult to say that it is universal or scientific. Therefore, please avoid making your self-citation rate too high.

It seems that you did not all of the corrections in figures that I pointed out before have been finished, but is it possible to align the font, line thickness, and character position?

Response 4 is little bit strange. 
